# Is It All a Conspiracy? Conspiracy Theories and People’s Attitude to COVID-19 Vaccination

**DOI:** 10.3390/vaccines9101051

**Published:** 2021-09-22

**Authors:** Zheng Yang, Xi Luo, Hepeng Jia

**Affiliations:** School of Communication, Soochow University, Suzhou 215127, China; zyang68@sheffield.ac.uk (Z.Y.); hpjia@suda.edu.cn (H.J.)

**Keywords:** COVID-19, vaccination, conspiracy theories, vaccine literacy, scientific literacy, construal level theory

## Abstract

A large body of research has found that people’s beliefs in conspiracy theories about infectious diseases negatively impacts their health behaviors concerning vaccination. Conspiracy belief-based vaccination hesitancy has become more rampant after the global outbreak of COVID-19. However, some important questions remain unanswered. For instance, do different versions of conspiracy theories—particularly conspiracy theories about the origin of the epidemic (e.g., that the SARS-CoV-2 leaked from a Wuhan virology laboratory or that the virus was of foreign origin) and the general theories about vaccine conspiracies (e.g., pharmaceutical companies covered up the danger of vaccines or people are being deceived about the effectiveness of vaccines)—have the same effect on vaccination intentions? Through a national survey adopting quota sampling in China, the current study tested the relationship between people’s conspiracy beliefs and their intention to receive the COVID-19 vaccination. The findings show that people’s embrace of conspiracy theories did indeed affect their intention to take COVID-19 shots. However, only conspiracy theories related to vaccines had a significant impact, while belief in more general theories about COVID-19 did not significantly affect vaccination intentions. People’s knowledge of vaccines (vaccine literacy) played an important role in this relationship. People with lower beliefs in vaccines conspiracy theories and higher levels of vaccine literacy were more likely to receive the COVID-19 vaccination.

## 1. Introduction

Since the end of 2019, the COVID-19 pandemic has ravaged the world and developed into the most serious public health issue affecting every country around the globe [1]. Long-term control of COVID-19 will depend on the development and effective public uptake of preventative vaccines [2]. However, according to data released by World Health Organization (WHO), as of 5 August 2021, when this study began, only 15.1% of the world’s population had been fully vaccinated. In many countries, boycotting and rejecting COVID-19 vaccines and vaccine hesitancy remain widespread [2,3,4,5]. Even in China, considered to be one of the countries with the smoothest vaccination program [6], as of 5 August 2021, China had only reported a total of 223 million people fully vaccinated against COVID-19 across the country [7], which is far less than the total population of China (1.41 billion). Many people in China are still reluctant and hesitant to be vaccinated, as evidenced by widespread social media posts showing public concern about the vaccine [8,9].

Therefore, improving effective communication around vaccination and further increasing the vaccination rate are key to the current response to the COVID-19 pandemic [2,10]. Among factors underlying people’s resistance to the COVID-19 vaccination, belief in conspiracy theories has been found to significantly negatively impact their preventative health behaviors or vaccination intentions [11,12,13,14,15]. People who believe in conspiracy theories may resist the authority of scientific or health experts who propose preventative measures and vaccination. Hence, they will be less compliant with professional public health guidance and suggestions [16,17]. The cognition and belief in conspiracy theories may also trigger negative public emotions, resulting in vaccine hesitancy and a decline in vaccine uptake [2,14,18].

However, not all studies have concluded that conspiracy beliefs reduce people’s intention to take preventive health measures and vaccinations. Marinthe and colleagues found that, at the early stages of the pandemic, people high in conspiracy mentality were more likely to engage in non-normative prevention behaviors, such as avoiding shaking hands and kissing, but were less willing to comply with extreme preventative behaviors that were government-driven [19]. Oleksy and colleagues further distinguished two different types of conspiracy theories during the pandemic—general conspiracy theories and government-related conspiracy theories—and found that only belief in government-related conspiracy theories had a significant correlation with fewer preventative measures, such as social distancing and handwashing [20].

The distinct health consequences of different types of conspiracy theories may be caused by the fact that various conspiracy beliefs induce mixed cognitive reactions [21]. Oleksy and colleagues observed that government-related conspiracy theories may have triggered the deprecation of public health policies and resistance to preventative measures [20]. Similarly, in Jia and Luo’s study, beliefs in non-COVID-19 conspiracies that blamed the West and embraced how SARS-CoV-2 (the virus that causes COVID-19) was of foreign origin both resulted in more active preventative measures in a Chinese sample when China largely controlled the pandemic [22]. It may be that belief in conspiracy theories in which foreign agents were the culprits increased the participants’ risk perception [22]. The timing of Jia and Luo’s study raised another issue—the temporal effect of conspiracy belief. Although Valensise and colleagues found that misinformation about COVID-19 did not dampen vaccine acceptance, there has been no systematic scholarly effort to explore the temporal effect of conspiracy beliefs [23]. Indeed, the temporal factor might overlap with the content of conspiracy theories, making some theories increase people’s risk perception at certain times but decrease it at others. 

Returning to the COVID-19 vaccination setting, there are two major types of conspiracy theories: COVID-19-related ones and general theories surrounding vaccines, which were spread long before the pandemic [13]. While studies have revealed that vaccine-related conspiracy theories significantly lowered people’s vaccination intentions [14,24,25], and some studies have revealed that COVID-19-related conspiracy theories negatively impacted people’s preventative health behaviors or vaccination intentions [11,12,13,14,15], the question remains whether COVID-19 conspiracy theories and vaccine theories have the same effect on influencing people’s intention for COVID-19 vaccination. This is the first focus of this study.

While conspiracy beliefs may not lead to a universal health consequence [11,12,13,14,15,19,22], studies show that the public’s issue-specific knowledge level and more generalized scientific literacy can generally reduce the negative consequences of such belief in vaccine uptake [14,15,24,26,27,28]. Since the communication of conspiracy theories is often based on the spread of incorrect or insufficient information [29], people with higher scientific literacy were found to be less likely to believe conspiracy theories [30]. Issue-specific knowledge has been found to protect people from a negative impact of conspiracy theories on their vaccination intention, while scientific literacy can slightly moderate the effect of COVID-19 conspiracy theories on people’s preventative behaviors [14,22,24]. Considering the above reasoning, we may legitimately ask whether scientific literacy and vaccine knowledge can adjust people’s intention for COVID-19 vaccination, which may be negatively impacted by different conspiracy theories.

To examine the above questions, we performed a survey based on a national quota sampling in China. With the global spread of COVID-19, many conspiracy theories are dominant in China. These include both COVID-19-related conspiracy theories (such as SARS-CoV-2 being leaked from a Wuhan virology laboratory or that the virus was of foreign origin [31]) and vaccine-related theories (such as pharmaceutical companies covered up the danger of vaccines or people are being deceived about the effectiveness of vaccines [32]).

## 2. Materials and Methods

### 2.1. Study Design

This study was based on an anonymous, self-designed, and structured nationwide online questionnaire survey, which was administered during April 2021 by the Shanghai-based survey firm Diaoyanba. The sample pool of the company covered 31 provincial administrative units in China. The questionnaire stressed anonymity and protection of privacy and allowed participants to exit at any time if they felt uncomfortable. During the COVID-19 pandemic, the method of online questionnaire survey has been widely used in epidemiology, public health management, medical sociology, and other related fields, as it can help to quickly obtain structural social and behavioral data about the epidemic when offline investigations are unavailable [33,34,35].

The online questionnaire employed in this study was aimed at assessing the following four points:(1)The Chinese public’s attitude toward COVID-19 vaccines and their intention to get vaccinated.(2)The Chinese public’s perceptions of COVID-19 vaccinations in China (including perception of risk, effectiveness, safety, and accessibility perception).(3)The Chinese public’s knowledge about COVID-19 vaccinations and their scientific literacy.(4)The Chinese public’s awareness of and belief in conspiracy theories.

The demographic data captured for each surveyed individual included age, gender, education level, monthly income, and home province. In addition, data about whether the surveyed individual was already vaccinated (at least one dose) was used to help to select samples suitable for further analysis.

The survey was conducted between 1 and 8 April 2021. When the survey was released, China’s COVID-19 pandemic had been controlled to a certain extent, with the average number of new infections in 7 days at just 39, all among people returning from overseas. Due to the tiny proportion of confirmed infections among the huge Chinese population, we did not consider being infected as a criterion for respondent enrollment.

Starting in January 2021, China’s first batch of COVID-19 vaccines, SARS-CoV-2 Inactivated Vaccine (Vero Cell, Beijing, China), began to be administered; by 1 April, China reported a total of 11.821 million doses of SARS-CoV-2 Inactivated Vaccine (Vero Cell) across the country [7]. In the face of China’s huge population, during the survey period, the unvaccinated still occupied the vast majority of China’s total population. Further promotion of vaccinations remains the focus of China’s current efforts to combat COVID-19. The questionnaire was pretested by the researchers. The final Cronbach’s alpha of the survey was 0.917.

### 2.2. Statistical Analysis

Statistical analysis was performed using SPSS Version 27.0 (IBM Corp, Armonk, NY, USA). When describing the Chinese public’s intention to vaccinate and their perception of conspiracy theories, the Descriptive Statistics Frequency function of SPSS was applied. The relationship between the Chinese public’s intention to vaccinate and their understanding of conspiracy theories and scientific knowledge level was analyzed with the hierarchical linear regression function of SPSS. A *p*-value < 0.05 was considered statistically significant, and the exact values were reported in the text unless *p* < 0.001.

## 3. Results

### 3.1. Demographic Characteristics

The survey was completed by 2038 individuals, of which 1890 (92.7%) had not received any COVID-19 vaccination before the survey, who were used in further analysis to better focus on the public intention to receive COVID-19 vaccinations. The demographic breakdown of the studied group is presented in Table 1. Most of those surveyed were middle aged (40–49, 26.9%; 30–39, 26.2%), were male (51%), had completed junior high school education (34.6%) and undergraduate education (32.8%), and earned 3001–5000 CNY (36.8%) or 5001–10,000 CNY (29.5%) per month.

### 3.2. Intention to Get the COVID-19 Vaccine

The two following questions were design to test the Chinese public’s intention to get vaccinated for COVID-19, with six corresponding answer options: “If you have not received the COVID-19 vaccine, please make your judgement on the following statements” and “If you plan to receive the COVID-19 vaccine and can decide when to get it for yourself, please make your judgement on the following statements”. A seven-point Likert-type scale was selected to measure the level of the Chinese public’s self-awareness of their intention to get the vaccine (Table 2). The survey data show that the Chinese public generally had a more positive attitude toward COVID-19 vaccination. In fact, 74.8% of respondents believed to varying degrees that they were likely to get COVID-19 vaccines this year; among the whole sample, most (34.9%) totally agreed that they were likely to get COVID-19 vaccines this year. Furthermore, 76.5% of respondents believed to varying degrees that they planned to get COVID-19 vaccines this year. However, some still had doubts about vaccination: 58.3% of respondents were not very inclined to think that they would get COVID-19 vaccines immediately. Most respondents (63.6%) thought to varying degrees that they planned to get COVID-19 vaccines within 1 month, whereas 30.7% of respondents still thought to varying degrees that they planned to wait and see before deciding when to get COVID-19 vaccines. These data show that most of the Chinese public generally had a more positive attitude toward COVID-19 vaccination, while a significant proportion held a more conservative and hesitant attitude. 

### 3.3. Conspiracy Beliefs

The survey distinguished the public’s perceptions of COVID-19 conspiracy theories—such as Cov-SARS-2 originating from a virus research institute in Wuhan, or 5G technology helping its spread—from the perception of vaccine-related conspiracy theories—such as data on the effectiveness of COVID-19 vaccines being fabricated by pharmaceutical companies, or that vaccinating children is harmful. In the survey, 15 questions were designed to measure Chinese public awareness and recognition of conspiracy theories, of which six were related to the COVID-19 pandemic and the virus (COVID-19 conspiracy theory beliefs, mainly referring to research by Jia and Luo) [22,30], five were related to vaccines (vaccine-related conspiracy theories, mainly coming from the WHO standard scale), and four were related to other political topics (other conspiracy theories). Five-point Likert-type scales were selected to measure the level of the Chinese public’s acceptance of those conspiracy theories (Table 3). It should be noted that all the conspiracy theories used in this study, such as “the Cov-SARS-2 originated from a virus research institute in Wuhan” are not scientifically proven, but only used in academic investigations to help researchers to understand and compare the Chinese public’s cognition of different types of conspiracy theories.

The survey data show that the Chinese public had a generally rational perception of different types of conspiracy theories. For the 11 different types of conspiracy theories, which had an average Cronbach’s alpha of 0.932, an average of 28.9% of respondents showed different types of disbelief. In the face of the COVID-19 pandemic, compared with vaccine conspiracy beliefs, an average of 17.3% of respondents showed different types of beliefs; the Chinese public were more inclined to believe COVID-19 conspiracy theories (19.8%). In particular, for the conspiracy theory that “the type of Cov-SARS-2 in the United States appeared earlier, indicating that the United States is more likely to be the source of the virus”, 41.7% of respondents showed different levels of belief in this conspiracy theory. However, for the conspiracy theory that “the Cov-SARS-2 originated from a virus research institute in Wuhan”, only 5.4% respondents tended to agree. Luo and Jia found that the patriotism of the Chinese public has a significant moderating effect on their perception of conspiracy theories and further anti-epidemic behaviors. For those conspiracy theories that treat China as a culprit and are not conducive to China’s national image, the Chinese public is more inclined to refuse to believe them [36]. However, the data also show that a certain proportion of the Chinese public does believe in conspiracy theories related to the COVID-19 pandemic, the Cov-SARS-2, and vaccines. The Chinese public’s awareness and acceptance of such conspiracy theories may have an impact on their intention to receive the COVID-19 vaccination.

### 3.4. Scientific Literacy and Vaccine Knowledge

Some studies have shown that people’s knowledge level may positively affect their health behavior and generally reduce the negative consequences of conspiracy theories belief in vaccine uptake [14,15,24,26,27,28]. During the pandemic, a higher level of knowledge is related to greater adoption of some epidemic prevention behaviors and vaccination uptake [24,25,26,27,28]. Similar to the classification of conspiracy theories mentioned above, this survey also investigated the basic scientific literacy and vaccine knowledge of the Chinese public. In the survey, 11 questions were designed to measure the Chinese public’s general scientific literacy, and six questions were designed to measure their vaccine knowledge. Most questions consisted of three options: the statement is ‘wrong’, the statement is ‘correct’, or ‘I do not know’ (Table 4). Selecting the right option was assigned a score of 1 and choosing the incorrect option or ‘I do not know’ was assigned a score of 0. According to Figure 1, the scientific literacy and vaccine knowledge of the Chinese public are generally at a relatively high level. Respondents with a score of 6 or above in scientific literacy level accounted for 58.9% (*n* = 1890), and respondents with a score of 3 or above in vaccine knowledge level accounted for 65.4% (*n* = 1890). This shows that the Chinese public surveyed had some basic understanding and knowledge of vaccines.

### 3.5. The Association among Conspiracy Theory Beliefs, Knowledge, and Intention to Get Vaccinated

After controlling for demographic variables (gender, age, education, and income), we used a hierarchical regression model to test the hierarchical correlation among the knowledge factor (scientific literacy and vaccine knowledge), conspiracy belief factors (COVID-19 conspiracy theory beliefs and vaccine conspiracy theory beliefs), and the Chinese public’s intention to get vaccinated. Hierarchical regression is a way to show if the untreated variables explain a statistically significant amount of variance in the dependent variable (DV) after accounting for all other variables [37,38], which is widely used in sociology, psychology, and other fields and is also applicable to this study. The survey data show that not all conspiracy theories were significantly associated with the Chinese public’s vaccination intentions (Table 4). Only vaccine conspiracy theory beliefs were shown to significantly negatively affect the Chinese public’s intention to receive COVID-19 vaccines, which means that the more the Chinese public believes in conspiracy theories related to vaccines, such as “the government often conceals the safety deficiencies of COVID-19 vaccines”, the less they are inclined to get the vaccine. COVID-19 conspiracy theory beliefs, which most studies believe may have a powerful impact on people’s vaccination intentions [36,39,40], did not show a significant correlation in this survey, regardless of whether they believed that COVID-19 originated in China (China as culprit) or abroad (foreign origin).

The survey data also shows that the level of knowledge the public has can adjust the negative impact of the Chinese public’s conspiracy theory beliefs on their COVID-19 vaccination intention. However, like the public’s beliefs in conspiracy theories, only the Chinese public’s knowledge related to vaccines has a positive effect on their intention to vaccinate, which means that the more knowledge the Chinese public has about COVID-19 vaccines, the greater their intention to receive the COVID-19 vaccine. But general science literacy is not significantly related to the Chinese public’s intention to get the COVID-19 vaccine (Table 5).

## 4. Discussion and Conclusions

The findings offer further support for exploring the relationship between the Chinese public’s intention to receive COVID-19 vaccines and their conspiracy theory beliefs. The results are different from many studies claiming that conspiracy beliefs can generally predict the public’s vaccine hesitancy and also negatively affect the public’s intention to vaccinate [36,41,42,43]. This study shows that different kinds of conspiracy theories have distinctive impacts; only belief in the conspiracy theories relevant to vaccines (not just the COVID-19 vaccine) had a significant negative impact on the Chinese public’s intentions to get vaccinated, whereas believing conspiracy theories related to the pandemic was not significantly relevant to the Chinese public’s intention to get vaccinated, regardless of whether they treated China or foreign countries as the ‘culprits’. Similarly, knowledge level increased vaccination intention, but the effect was only limited to vaccine knowledge, which had a significantly positive impact on the public vaccination intention; their scientific literacy did not.

This may be explained using the construal level theory, which points out that individuals have different interpretations of different events and beliefs. These different interpretations also have different degrees of impact on individuals. These different levels of interpretation and influence depend on the psychological distance of cognitive objects perceived by people. That is, when individuals perceive that the psychological distance between the object event or belief and themselves or their target behavior is great, the object event or belief has less impact on them and their target behavior [44,45]. In this study, the vaccine conspiracy theory beliefs and vaccine-related knowledge were closer to the Chinese public’s target behavior (COVID-19 vaccination) at the psychological level, while the COVID-19 conspiracy theory beliefs and scientific literacy were further away. Therefore, this study shows that only vaccine conspiracy beliefs and vaccine-related knowledge had a significant impact on the Chinese public’s intention to obtain a COVID-19 vaccination.

This study first suggests that, when analyzing the health consequences of conspiracy beliefs or other cognitive factors, such as their impacts on the public’s intention to get vaccinated, we cannot simply regard conspiracy theories as a homogenous mass, even though conspiracy beliefs are widely known to discourage vaccination [16,17]. We need to divide them according to specific contexts, and then analyze the impact of different types of conspiracy theories on the public’s intention to get vaccinated. This may not only appear in the specific Chinese context but may also be of some significance for the international environment. For example, Oleksy and colleagues also found that belief in government-related conspiracy theories had a significant correlation with fewer preventative measures such as social distancing and handwashing in the Western context [20].

Second, many studies have demonstrated that COVID-19 conspiracy beliefs also had a significant negative impact on the public’s intention and behavior to take preventative measures or get vaccinated against the virus in their countries at the early stage of the pandemic [36,41,42,43]. This also suggests that different temporal and sociocultural backgrounds may mediate the impact of conspiracy theory beliefs on the public’s intentions regarding vaccination. Therefore, when exploring the link between conspiracy beliefs and vaccination, we need to abandon the global universal and Western-centric thinking and base research on the local specific context. This may be applicable for different countries when thinking about their own vaccination work. For China, to enhance the public’s intention to get vaccinated, it will be more effective to refute the rumors of vaccine conspiracy theories than COVID-19 and other conspiracy theories [22]. However, this conclusion may not be applicable in other periods of pandemic and other sociocultural contexts. Valensise and colleagues further pointed out that we may have exaggerated the impact of some information factors on the public’s vaccination intentions and their vaccine acceptance rates [23]. This also reminds us that maybe not all conspiracy theories have a significant impact on the public’s vaccination intentions and their vaccine acceptance rates.

This study is not without limitations and invites extensions. Firstly, this study was mainly based on the Chinese context. Therefore, the main conclusion of our study, that different kinds of conspiracy theories have different effects on people’s intention to get the COVID-19 vaccine, may particularity apply to this study context. Situations may differ as a function of cultural environment. Secondly, although the survey was cross-sectional and nationally representative, the sample size was small compared with China’s huge population. Further studies on other cognitive variables which may also contribute to the public’s intention to receive vaccines were not considered in this study, and follow-up research can choose to address this limitation.

## Figures and Tables

**Figure 1 vaccines-09-01051-f001:**
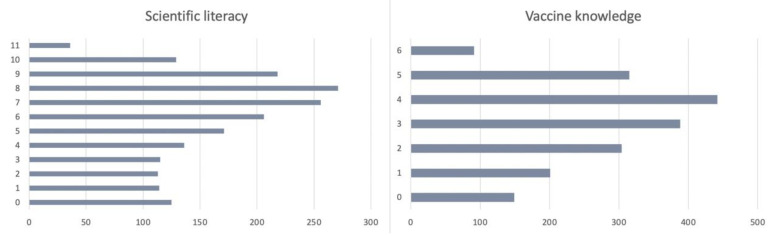
Level of scientific literacy and vaccine knowledge of participants (*n* = 1890).

**Table 1 vaccines-09-01051-t001:** Distribution of demographic characteristics of the sample (*n* = 1890).

Variable	% (*n*)
**Age**	
18–29	22.3 (421)
30–39	26.2 (496)
40–49	26.9 (509)
50–59	24.6 (464)
**Gender**	
Male	51.0 (964)
Female	49.0 (926)
**Education Level**	
Junior high school and below	12.7 (240)
Senior high school	17.8 (336)
Junior college education	34.6 (654)
College degree	32.8 (620)
Postgraduate and above	2.1 (40)
**Monthly Income**	
3000 or less	26.1 (493)
3001–5000	36.8 (696)
5001–10,000	29.5 (558)
10,001–20,000	6.3 (120)
More than 20,000	1.2 (23)

**Table 2 vaccines-09-01051-t002:** The Chinese public’s intention to get the COVID-19 vaccine (*n* = 1890).

Question: If You Have Not Received the COVID-19 Vaccine, Please Make your Judgement on the Following Statements:							
	Totally Disagree	Disagree	A Little Disagree	Neutral	A Little Agree	Agree	Totally Agree
I will probably get the COVID-19 vaccines this year.	1.2% (23)	1.2% (22)	1.6% (30)	21.2% (401)	12.3% (233)	27.6% (521)	34.9% (660)
I plan to get the COVID-19 vaccines this year.	1.3% (25)	1.0% (18)	1.5% (29)	19.6% (371)	10.7% (203)	29.8% (564)	36.0% (680)
**Question: If You Plan to Receive the COVID-19 Vaccine and Can Decide When to Get It for Yourself, Please Make Your Judgement on the Following Statements:**							
I plan to get the COVID-19 vaccines immediately.	4.1% (78)	2.6% (50)	3.5% (67)	48.1% (909)	9.5% (180)	21.6% (408)	10.5% (198)
I plan to get the COVID-19 vaccines within 1 month.	4.3% (82)	3.5% (67)	5.3% (101)	23.2% (438)	11.1% (209)	20.7% (392)	31.8% (601)
I plan to get the COVID-19 vaccines within 3 months.	8.9% (168)	6.5% (123)	6.8% (129)	42.0% (793)	15.2% (287)	17.4% (328)	3.3% (62)
I plan to wait and see before deciding when to get the COVID-19 vaccines.	9.3% (176)	11.5% (217)	8.3% (157)	40.3% (761)	11.0% (207)	14.0% (265)	5.7% (107)

**Table 3 vaccines-09-01051-t003:** Participants’ belief in conspiracy theories (*n* = 1890).

Question: To What Extent Do You Agree with the Following Statements?					
	Totally Disagree	Disagree	Neutral	Agree	Totally Agree
**COVID-19 Conspiracy Theory Beliefs**					
COVID-19 vaccine effectiveness data are often fabricated by pharmaceutical companies.	41.5% (784)	17.0% (322)	22.0% (416)	10.3% (195)	9.2% (173)
The Cov-SARS-2 came out of foreign military laboratories.	31.1% (587)	13.2% (249)	35.8% (676)	9.2% (173)	10.8% (205)
The Cov-SARS-2 originated from a virus research institute in Wuhan.	59.6% (1127)	13.4% (254)	21.5% (406)	4.4% (84)	1.0% (19)
In October 2019, foreign athletes infected with the Cov-SARS-2 came to Wuhan to participate in the Military Games, which led to the spread of new crown pneumonia.	33.3% (630)	13.6% (257)	34.2% (647)	8.8% (167)	10.0% (189)
5G technology helps spread the Cov-SARS-2.	61.0% (1153)	9.4% (177)	14.1% (267)	7.4% (139)	8.1% (154)
The type of Cov-SARS-2 in the United States appeared earlier, indicating that the United States is more likely to be the source of the virus.	25.0% (473)	11.2% (212)	22.1% (418)	15.6% (294)	26.1% (493)
**Vaccine-Related Conspiracy Theories**					
Vaccine’s effectiveness data are often fabricated by pharmaceutical companies.	41.5% (784)	17.0% (322)	22.0% (416)	10.3% (195)	9.2% (173)
The government often conceals the safety deficiencies of vaccines.	43.0% (813)	18.1% (342)	22.3% (421)	9.3% (175)	7.4% (139)
Pharmaceutical companies cover up the danger of vaccines.	42.3% (817)	17.4% (329)	22.0% (415)	9.2% (174)	8.2% (155)
People are often deceived about the effectiveness of the vaccines.	44.1% (834)	15.2% (288)	21.7% (410)	9.7% (184)	9.2% (174)
The fact that vaccines are harmful for children is deliberately obscured.	54.1% (1023)	13.3% (252)	18.3% (346)	6.2% (118)	8.0% (151)

**Table 4 vaccines-09-01051-t004:** Participants’ scientific literacy and knowledge about vaccine (*n* = 1890).

General Scientific Questions			
**Question: Two Scientists Want to Know Whether a Drug for High Blood Pressure Is Effective. The First Scientist Distributed the Drug to 1000 Patients with Hypertension, and then Observed How Many Patients Had Their Blood Pressure Decreased. The Second Scientist Divided the Patients into Two Groups. The First Group of 500 Patients with Hypertension Took Medicine, While the Other Group of 500 Patients Did Not Take Medicine. Then, He Observed How the Blood Pressure Decreased in the Two Groups. Which of the Two Scientists Is More Effective in Testing the Effect of Drugs?**	**First One**	**Second One**	**I Do Not Know**
	14.2% (269)	69.9% (1322)	15.8% (299)
Question: The doctor told a couple that, because they have the same morbid genes, if they give birth to a child, the child’s chance of genetic disease is 1 in 4. Do you think the following statement is correct?	Wrong	Correct	I do Not know
If they have three children, none of them will get genetic diseases.	61.5% (1162)	8.3% (156)	30.2% (572)
If their first child has a genetic disease, the next three children will not have a genetic disease.	61.6% (1165)	9.1% (171)	29.3% (554)
If the first three children are healthy, the fourth child must have a genetic disease.	59.5% (1124)	8.9% (169)	31.6% (597)
Their children may have genetic diseases.	14.8% (280)	56.8% (1073)	28.4% (537)
**Question: Do You Think the Following Statement Is Correct?**			
Cov-SARS-2 can cause SARS and pneumonia, but it will not cause colds.	55.2% (1042)	10.0% (190)	34.8% (658)
Electrons are smaller than atoms.	23.0% (435)	31.0% (585)	46.0% (870)
The mother’s genes determine whether the child is a boy or a girl.	71.4% (1350)	7.6% (144)	21.0% (396)
Lasers are produced by converging sound waves.	22.2% (419)	17.8% (336)	60.0% (1135)
Antibiotics (such as penicillin, streptomycin, or cephalosporin) can kill both bacteria and viruses.	34.5% (746)	24.9% (471)	35.6% (673)
If you eat genetically modified fruit, human genes may change.	49.7% (940)	13.4% (291)	34.9% (659)
**Vaccine-Related Questions (Vaccine Literacy)**			
We do not necessarily need a vaccine because diseases can always be cured.	77.2% (1460)	2.3% (44)	20.4% (386)
Smallpox will not be eradicated unless vaccines are widely used.	15.8% (298)	57.1% (1080)	27.1% (512)
If many vaccines are given too early, children’s immune system will not develop normally.	44.7% (844)	16.7% (315)	38.6% (731)
Vaccination does not increase the incidence of allergies.	36.1% (682)	15.4% (291)	48.5% (917)
If children are not vaccinated so much, they will be more resistant to disease.	61.6% (1165)	9.3% (176)	29.1% (549)
Autism, multiple sclerosis, and diabetes may be caused by vaccination.	54.1% (1022)	8.1% (154)	37.8% (714)

**Table 5 vaccines-09-01051-t005:** Hierarchical regression model of factors associated with intention to receive COVID-19 vaccines (*n* = 1890).

Variable	Intention to COVID-19 Vaccination		
	Model 1	Model 2	Model 3
	*p*	*p*	*p*
**Demographic Factors**			
Gender (ref. female) Male	−0.039	−0.032	−0.023
Age	0.033	0.040	0.039
Education (ref. primary/lower secondary)			
Upper secondary	0.052	0.017	0.019
Junior college	0.088 *	0.061	0.061
Undergraduate degree	0.152 ***	0.079 *	0.077 *
Postgraduate degree	−0.003	−0.023	−0.023
Income	0.059 *	0.038	0.041
**Attitude**		0.260 ***	0.249 ***
**Knowledge Factors**			
Science knowledge		0.037	0.012
Vaccine knowledge		0.196 ***	0.177 ***
**Conspiracy Belief Factors**			
COVID-19 conspiracy theory beliefs: COVID-19’s foreign origin			0.038
COVID-19 conspiracy theory beliefs: China as culprit			−0.009
Vaccine conspiracy theory beliefs			−0.118 **
**Model Statistics**			
Adjusted *R*^2^	0.016	0.144	0.154
Δ*R*^2^	0.020	0.129	0.011
Δ*F*	5.469 ***	94.778 ***	8.114 ***
Model *F*	5.469 ***	32.834 ***	27.416 ***

* *p* < 0.05, ** *p* < 0.01, *** *p* < 0.001.

## Data Availability

Materials and anonymous data are available from the authors upon request.

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
