# Peer review of "Is It All a Conspiracy? Conspiracy Theories and People’s Attitude to COVID-19 Vaccination"

_vaccines, 2021, doi:10.3390/vaccines9101051_

Round 1
Reviewer 1 Report
- What type(s) of vaccine(s) (mRNA, vector, inactivated…) will be administered to the people in the study population in China? Detailed information on this would be helpful for understanding of readers around the world.
- Table 2 ‘I plan to get the COVID-19 vaccines in three months’, the sum of respondent number is 1,880≠1,890.
- Did the investigators take past medical history of COVID-19 in the study population into consideration?
Author Response
We gratefully thank the editors and all reviewers for their time spent making their constructive remarks and useful suggestions, which have significantly raised the quality of this manuscript and has enabled us to improve it. Each suggested revision and comment put forward by the reviewers was incorporated and considered. Below the comments of the reviewers are responded to point-by-point and the revisions to the text have been indicated. We have had to be concise in our revisions due to the word count but are more than happy to add any more detail should this be required.
Reviewer 1:
Comment 1: What type(s) of vaccine(s) (mRNA, vector, inactivated…) will be administered to the people in the study population in China? Detailed information on this would be helpful for understanding of readers around the world.
Reply: Many thanks for your comment and suggestion. The type of the vaccines, SARS-CoV-2 Inactivated Vaccine (Vero Cell), has been added in the revised manuscript. We hope this information could be helpful for the understanding of readers around the world.
Comment 2: Table 2 ‘I plan to get the COVID-19 vaccines in three months’, the sum of respondent number is 1,880≠1,890.
Reply: We are very sorry for this error in data transcription. We have corrected the data on Table 2. Many thanks for your comment.
Comment 3: Did the investigators take past medical history of COVID-19 in the study population into consideration?
Reply: Many thanks for your question. Indeed, we did not take the medical history of COVID-19 in this study, because COVID-19 has been very effective in China since its’ breakout, the infection rate is very small comparing the whole population of China. During the survey, the average number of infected people in 7 days was only 39. Therefore, we did not take medical history as our main consideration.
Reviewer 2 Report
Dear Authors,
your study is really interesting. It's well written and clear.
The introduction section properly illustrates the topic. The material and method section is well written and explains the research approach. The results section provides important information to the reader. Discussion and conclusion are interesting. If it's possible, I suggest to add some information about the role of the theory at the international level.
Author Response
We gratefully thank the editors and all reviewers for their time spent making their constructive remarks and useful suggestions, which have significantly raised the quality of this manuscript and has enabled us to improve it. Each suggested revision and comment put forward by the reviewers was incorporated and considered. Below the comments of the reviewers are responded to point-by-point and the revisions to the text have been indicated. We have had to be concise in our revisions due to the word count but are more than happy to add any more detail should this be required.
Comment: your study is really interesting. It's well written and clear. The introduction section properly illustrates the topic. The material and method section is well written and explains the research approach. The results section provides important information to the reader. Discussion and conclusion are interesting. If it's possible, I suggest to add some information about the role of the theory at the international level.
Reply: Many thanks for your recognition and appreciation, and also the suggestion. We have tried to expand the significance of our research to the international scope in the final discussion part. We hope this could be more valuable. And we have further sought professional proofreading service to help to improve the English language and writing style. We hope this version could be better.
Round 2
Reviewer 1 Report
(There are no comments)